# In-silico study of accuracy and precision of left-ventricular strain quantification from 3D tagged MRI

Ezgi Berberoğlu[1], Christian T. Stoeck[1], Philippe Moireau[2,3], Sebastian Kozerke[1], Martin Genet[2,3]*

1 Institute for Biomedical Engineering, University and ETH Zurich, Zurich, Switzerland, 2 MΞDISIM team, Inria, Palaiseau, France, 3 Laboratoire de Mécanique des Solides (LMS), École Polytechnique, C.N.R.S., Institut Polytechnique de Paris, Palaiseau, France

* martin.genet@polytechnique.edu

## Abstract

Cardiac Magnetic Resonance Imaging (MRI) allows quantifying myocardial tissue deformation and strain based on the tagging principle. In this work, we investigate accuracy and precision of strain quantification from synthetic 3D tagged MRI using equilibrated warping. To this end, synthetic biomechanical left-ventricular tagged MRI data with varying tag distance, spatial resolution and signal-to-noise ratio (SNR) were generated and processed to quantify errors in radial, circumferential and longitudinal strains relative to ground truth. Results reveal that radial strain is more sensitive to image resolution and noise than the other strain components. The study also shows robustness of quantifying circumferential and longitudinal strain in the presence of geometrical inconsistencies of 3D tagged data. In conclusion, our study points to the need for higher-resolution 3D tagged MRI than currently available in practice in order to achieve sufficient accuracy of radial strain quantification.

**Data Availability Statement:** The minimal dataset is publicly available at ETH Library, https://www.research-collection.ethz.ch/handle/20.500.11850/497459 (DOI: https://doi.org/10.3929/ethz-b-000497459).

## 1 Introduction

The noninvasive assessment of myocardial function represents an important diagnostic tool in the clinic. Beyond cine Magnetic Resonance Imaging (MRI), a number of dedicated approaches to quantify ventricular tissue motion and strains have been proposed, including phase contrast [1], tissue tagging [2], displacement encoding [3] and strain encoding [4].

In the present paper, we focus on three-dimensional (3D) tagged MRI [5], which is based on the principle of spatial modulation of magnetization [2, 6]. Since its inception, a number of improvements have been proposed [5, 7–10] and the method has been shown to reveal alterations in myocardial function for a variety of pathologies including ischemic heart disease [11, 12], aortic stenosis [13], cardiac hypertrophy [14], left bundle branch block [15], cardiomyopathy [16] and coronary artery disease [17], among others.

Processing techniques for tagged data can be divided into three main categories: direct, Fourier-based and tracking-based approaches [18]. While direct methods aim to extract tag features using image filtering and segmentation approaches [19], Fourier-based methods

**Funding:** French National Research Agency Research Grant Nr. ANR-10-EQPX-37 received by Martin Genet. • Swiss National Science Foundation (SNF) Research Grants Nr. CR23I3_166485 received by Sebastian Kozerke. Ezgi Berberoğlu's salary was paid by this grant. • Swiss National Science Foundation (SNF) Research Grants Nr. PZ00P2_174144 received by Christian T. Stoeck. His salary was also paid by the mentioned grant. The funders had no role in study design, data collection and analysis, decision to publish, or preparation of the manuscript.

**Competing interests:** The authors have declared that no competing interests exist.

exploit the Fourier-shift theorem by band-pass filtering a single tagging peak in k-space yielding a material point-specific image phase which can be tracked (HARmonic-Phase analysis (HARP) [20, 21]). Limitations include a reduction of spatial resolution due to band-pass filtering [22], as well as potential phase aliasing [23]. Using sine-wave modeling (SinMod), the noise sensitivity of HARP is partly addressed by providing improved accuracy of the displacement fields in low signal-to-noise ratio (SNR) scenarios [23]. SinMod was also demonstrated to perform well in 3D [24]. However, SinMod, like HARP and other phase-based approaches, requires sufficiently small displacements in-between temporal frames. Tracking-based approaches aim to find a deformation field to register any two images, typically involving also some form of regularization [18, 25]. Compared to Fourier-based approaches, which are applicable to tagged images only, tracking-based methods have a wider range of applications including cine, tagged [18, 26] or ultrasound images [27]. Among the tracking-based approaches, Finite-Element (FE)-based methods offer a convenient way to constrain the solution displacement field, with a proper discretization independent of the image discretization, thus naturally providing geometrical regularization [28–30]. Moreover, it allows incorporation of mechanical regularization, either based on basic mechanical principles, or using mechanical models [26]. These models require the prescription of appropriate boundary conditions and a material model to characterize the cardiac constitutive behavior.

Studies using MRI tagging have revealed regional differences in strain distribution [14, 31, 32]. While circumferential and longitudinal strains are reported frequently for both healthy subjects and patients, radial strain is often omitted due to limited accuracy and precision [33]. At the same time, radial strain is a decisive metric for assessing contractility of the heart [34].

It is the objective of the present work to analyze the dependency of radial, circumferential and longitudinal strain quantification from 3D tagged MRI with regard to tag distance, image resolution and SNR. To do so, synthetic tagged images are generated using a reference biomechanical left-ventricular (LV) model [35, 36] as input. Strain is then quantified from the synthetic images using a recently proposed FE-based method with mechanical regularization referred to as equilibrated warping [26].

## 2 Methods

Fig 1 summarizes the pipeline of this study, which includes the reference LV model (Fig 1a), rasterization, resampling and convolution (Fig 1b), image processing (Fig 1c) and image registration (Fig 1d).

### 2.1 Cardiac biomechanical model and reference left-ventricular motion

In this study, we utilize a generic anatomical model of the LV represented by a truncated ellipsoid with an approximate end-diastolic volume of 202 ml, decreasing down to 62 ml at end-systole (ES). The ventricular wall thickness is almost constant across the ventricle with approximate values of 15 mm and 22 mm at end-diastole (ED) and ES, respectively. We consider a reference biomechanical model of the LV [35, 37], which combines a passive visco-hyperelastic behavior with a micro-macro model of the active muscle contraction [37], and is coupled to a lumped cardiovascular circulation model [35]. FE resolution of the model provides the displacement field over the LV (from which one can compute any deformation metric, e.g., strain, strain rate, etc.) with 1000 time steps throughout the cardiac cycle. The model is used as ground truth in the study. To ensure a physiological behavior and prevent unrealistically large displacement velocities at the base of the LV [38], the displacement was interpolated in time, capping the maximum velocity to 0.1mm/ms (Fig 1a).

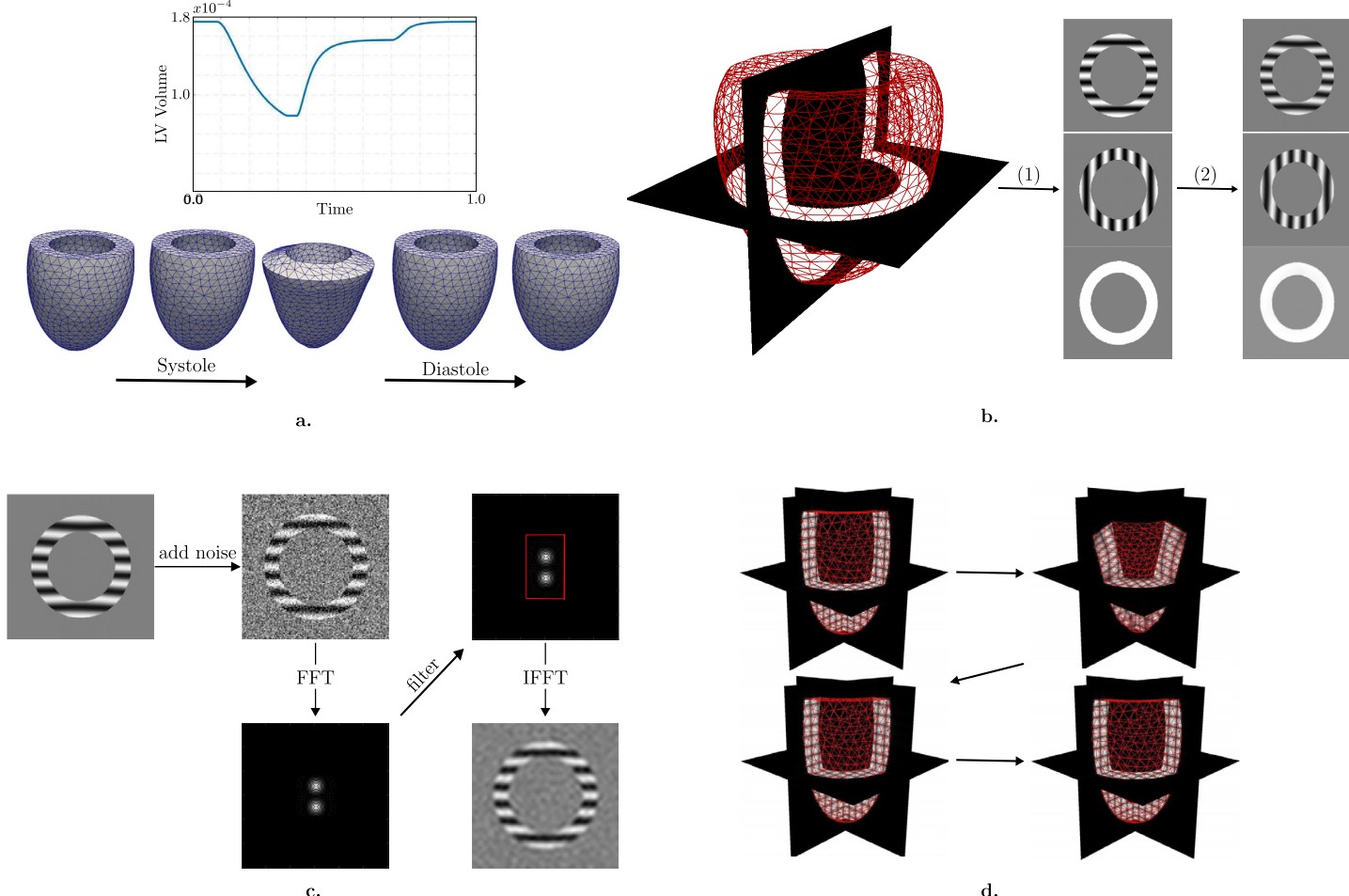

**Fig 1. Study design.** Reference biomechanical LV model simulating the full cardiac cycle (a). Synthetic image generation using the model (b). Three different image stacks with tag lines in orthogonal directions are generated by rasterization (1) and then resampled by cropping in k-space and convolved in time (2). Adding noise and changing image resolution by filtering in k-space (c). Image registration showing 3D tagged images superimposed with the warped mesh for different time frames (d).

## 2.2 Synthetic image generation

The modified LV model is further utilized to generate synthetic 3D-tagged images as follows. First, tagged images with 0.5 mm pixel size are generated from the time-resolved LV meshes through rasterization [39], and using a simplified complementary spatial modulation model [7] for 3D tagging without tag-line fading (Fig 1b-1). We assume a standard image generating function used in CSPAMM to create the tagged image stacks:

$$I = \sin\left(\frac{\pi X_i}{s}\right), \tag{1}$$

where $X_i$ corresponds to the spatial coordinate of the image voxel in the $i^{th}$ direction, and $s$ is the tagging distance, ranging from 3 mm to 7 mm (Fig 2a). Compared to other tagging techniques, in which the tagging pattern is approximated by higher order functions, (e.g., SPAMM) [7], here we assume the standard modulation pattern for CSPAMM as a "worst case" scenario in terms of tag line contrast. Considering the approximate ventricular wall thickness at ED, the synthetic images have between ca. 2 (for 7 mm tagging) and 5 (for 3 mm tagging) tag lines within the thickness of the myocardium.

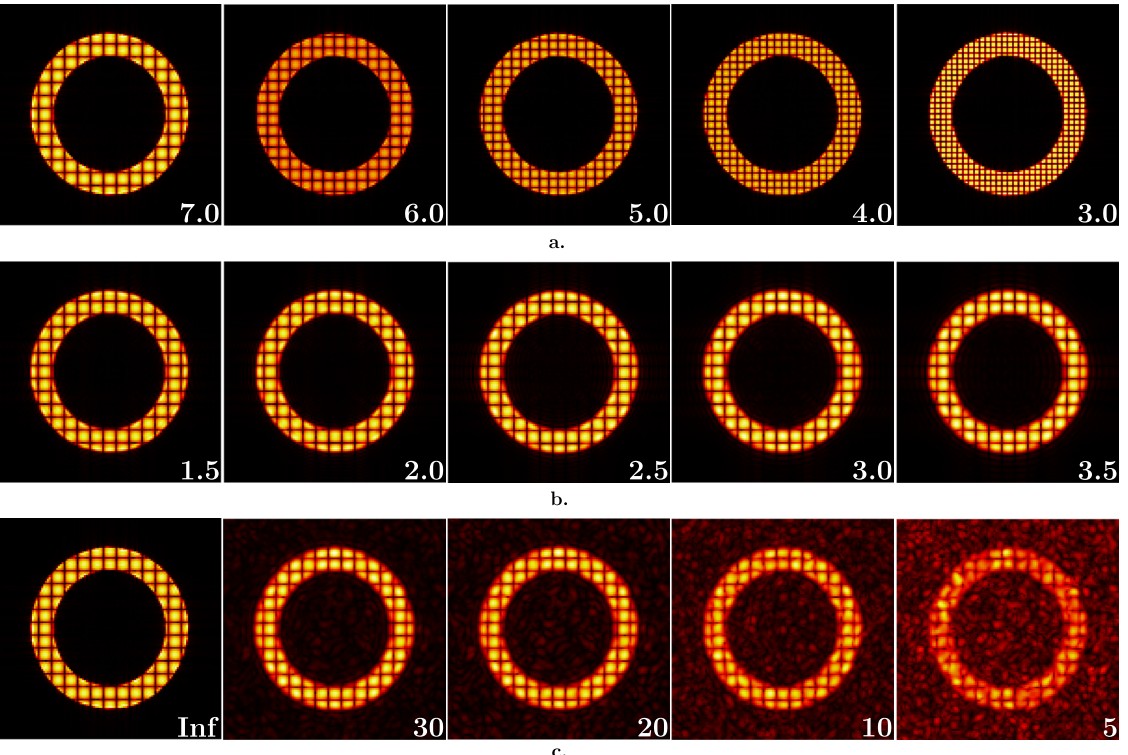

**Fig 2. Isotropic synthetic images to analyze the effect of image properties.** Images with varying tag distances (in mm) (a), pixel sizes (in mm) (b), and SNRs (c) (values are written at the bottom right corner for each case). Images shown in (a) are noiseless and have 1 mm pixel size. Images in (b) are noiseless and have 7 mm tag distance. For the ones in (c), pixel size and tag distance are 3.5 mm and 7 mm, respectively. Images represent the configuration at end-diastolic time frame.

Three stacks are generated, with $i$ = 1, 2, 3 for each tagging direction. Thereafter, images are resampled to 1 mm pixel size using zero-filling after cropping in k-space and convolution in time (Fig 1b-2) using a convolution window size of 40 ms [5]. Then, the images are resampled in time to yield 100 time steps throughout the cardiac cycle. These three series of images are then combined using:

$$I_0(\mathbf{X}) = \sqrt[3]{\left|\sin\left(\frac{\pi X_1}{s}\right)\right| \cdot \left|\sin\left(\frac{\pi X_2}{s}\right)\right| \cdot \left|\sin\left(\frac{\pi X_3}{s}\right)\right|}. \tag{2}$$

to generate the reference dataset used for 3D validation of the FE-based image registration technique (Fig 1d).

In addition to the reference images, datasets with varying image properties are generated (Fig 1c). First, different levels of Gaussian noise with zero mean are added to generate sets of images with varying SNR, which is defined as the ratio of the magnitude of the signal to standard deviation of the noise:

$$\text{SNR} = \frac{\text{Max}_{\text{signal}}}{\text{Std}_{\text{noise}}}. \tag{3}$$

Then, image spatial resolution is changed by filtering the reference dataset in k-space using a box filter with different bandwidth (Fig 1c). We generated image sets varying in tag distance (Fig 2a), pixel size (Fig 2b), and SNR (Fig 2c).

## 2.3 FE-based image registration

FE-based image registration is then performed on the synthetic datasets to investigate the effect of image properties on deformation quantification (Fig 1d). A continuum formulation of the image registration problem is introduced and later discretized using the FE method. Considering the reference and current configurations of the object represented by the images, $\Omega_0$ and $\Omega_t$, related image intensity fields are denoted by $I_0$ and $I_t$. A deformation map $\Phi(\mathbf{X})$ is defined between these two configurations to map the reference points $\mathbf{X} \in \Omega_0$ onto their spatial counterparts $\mathbf{x} = \Phi(\mathbf{X}) = \mathbf{X} + \mathbf{U}(\mathbf{X}) \in \Omega_t$, where $\mathbf{U}$ is the displacement field. In the presence of image noise, this problem is ill-posed and requires to be formulated as a minimization problem. Hence, the problem of finding the displacement field can be formulated as

$$\text{find } \mathbf{U} = \underset{\{U\}}{\text{argmin}} = \{h(\boldsymbol{U}) = (1 - \beta)\Psi^{\text{im}}(\boldsymbol{U}) + \beta\Psi^{\text{reg}}(\boldsymbol{U})\}, \tag{4}$$

which aims to minimize the functional $h$ expressed in terms of regularization strength $\beta$. The image similarity metric, $\Psi^{\text{im}}$,

$$\Psi^{\text{im}}(\boldsymbol{U}) = \frac{1}{2}\|I_t \circ \boldsymbol{\Phi} - I_0\|^2_{L^2(\Omega_0)}, \tag{5}$$

and the regularization energy, $\Psi^{\text{reg}}$,

$$\Psi^{\text{reg}} = \sum_K \frac{1}{2}\|\mathbf{Div}(\mathbf{F} \cdot \mathbf{S})\|^2_{L^2(K)} + \sum_F \frac{1}{2h}[\![(\mathbf{F} \cdot \mathbf{S} \cdot \mathbf{N})]\!]^2_{L^2(F)}, \tag{6}$$

are weighted by factors $(1 - \beta)$ and $\beta$, respectively, to rescale the regularization strength to stay in the range [0, 1]. K, F and $\mathbf{N}$ represent the set of finite elements, the set of interior facets, and the facets normal, respectively. $\mathbf{S}$ and $\mathbf{F}$ are the second Piola-Kirchhoff stress tensor and transformation gradients, respectively, and h is a characteristic length of the elements; see [26] for more details. The novelty in the method is the regularization technique which is a continuum finite strain formulation of the equilibrium gap principle introduced in [40], readily discretizable with standard finite elements. Compared to other mechanical regularization techniques, equilibrated warping enforces only the basic principle of mechanical equilibrium, but does not constrain strain magnitude in any way. Moreover, the registration problem is geometrically regularized by the FE mesh. In addition to the equations specified above, Eq (6) requires the specification of a constitutive model, chosen here as Neohookean compressible hyperelastic potential with Ciarlet-Geymonat [41]:

$$\rho_0\psi = \frac{\kappa}{2}(J^2 - 1 - \ln(J)) + \frac{\mu}{2}(I_C - 3 - 2\,\ln(J)), \tag{7}$$

in terms of bulk and shear modulus, $\kappa$ and $\mu$, volume map, $J = \text{Det}(\mathbf{F})$, and $I_C = \text{Tr}(\mathbf{C})$. Here, $\mathbf{F} = \mathbf{Grad}(\Phi)$ and $\mathbf{C} = \mathbf{F}^T\mathbf{F}$ are the deformation gradient and right Cauchy-Green deformation tensor, respectively. Following the general FE procedure, the variational form of Eq (4) is obtained by derivation. It is then linearized, and discretized using standard continuous Lagrange elements. More details on the formulation and solution procedure can be found in [26]. The method implementation is freely available as a python library (https://gitlab.inria.fr/mgenet/dolfin_warp) implemented based on FEniCS [42] and VTK (http://www.vtk.org) libraries.

The impact of the regularization parameter and the mechanical model utilized are further discussed in [26, 43]. In this study, for each image set, we run the registration algorithm for a range of regularization strength $\beta$ and use the best performing value (i.e. estimated motion closest to ground truth) in the analysis. The reader is referred to [26] for a more detailed

discussion on the choice of $\beta$ in case the ground truth is unknown. By designing this "optimal" registration technique, we focus the study on the image content itself, not the image processing approach.

## 2.4 Performance metrics for cardiac strain quantification

Strain quantification performance is assessed through computing the normalized mean and standard deviation of the norm of the displacement error given by:

$$d_i = \frac{\|\mathbf{u}_{reg}^i - \mathbf{u}_{ref}^i\|}{\max\|\mathbf{u}_{ref}\|}, \tag{8}$$

$$d_{avg} = \frac{1}{N_n}\sum_{i=1}^{N_n} d_i \times 100(\%), \tag{9}$$

$$d_{std} = \frac{1}{N_n}\sum_{i=1}^{N_n}(d_i - d_{avg})^2 \times 100(\%), \tag{10}$$

where $d_{avg}$ and $d_{std}$ are the normalized mean and standard deviation in displacement error norm. $\mathbf{u}_{reg}^i$ and $\mathbf{u}_{ref}^i$ are the displacement vectors at node i at end-systolic time frame, for the registered case and ground truth, respectively. Moreover, $N_n$ is the total number of nodes.

In addition to the displacement error, we investigate the component-wise sensitivity of the strain to a change in image characteristics by computing the mean and standard deviation according to:

$$e_i = \mathbf{s}_{reg}^i - \mathbf{s}_{ref}^i \tag{11}$$

$$e_{avg} = \frac{1}{N_{el}}\sum_{i=1}^{N_{el}} e_i \tag{12}$$

$$e_{std} = \frac{1}{N_{el}}\sum_{i=1}^{N_{el}}(e_i - e_{avg})^2 \tag{13}$$

where $e_{avg}$ and $e_{std}$ are the mean and standard deviation in strain error for component s, which is a scalar field, while $\mathbf{s}_{reg}^i$ and $\mathbf{s}_{ref}^i$ are the strain values at element i at end-systolic time frame for the registered case and the ground truth, respectively.

In this study, we investigate both the individual and combined effects of two image characteristics, Tag distance to Pixel size Ratio (TPR) and SNR on images with isotropic (Fig 2) and anisotropic (Fig 3) spatial resolution. The first part, analysis on isotropic images, includes noiseless images with TPR ranging from 2.8 to 7.0 (Fig 2(a) and 2(b)), and images with varying levels of SNR (from 5 to 30) (Fig 2c), keeping the tag distance and pixel size constant: 7 mm and 1 mm, respectively. The combined effect of TPR and SNR is investigated on the image set with isotropic pixel size ranging from 1.5 mm to 3.5 mm having different SNRs (from 5 to 30) and a tag distance of 7 mm.

The second part is focused on the analysis of anisotropic image resolutions. Similarly, individual effects of resolution and SNR are analysed first. For this purpose, we generated noiseless images with 7 mm tag distance and pixel size ranging from 1.5 mm to 3.5 mm in the tagging direction, while the transverse directions are assigned to a larger pixel size ranging from 3.5

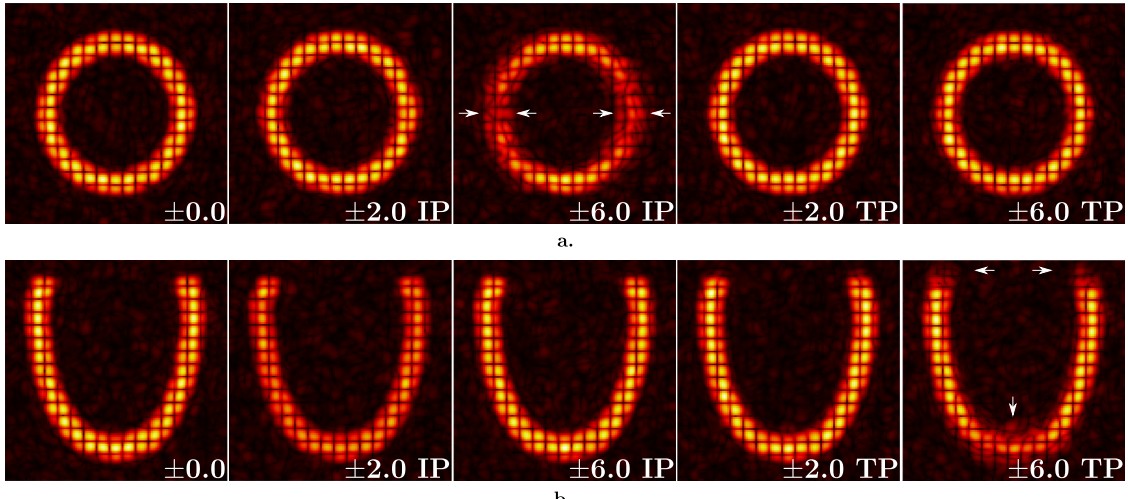

**Fig 3. Anisotropically resolved synthetic images to analyze the effect of geometrical inconsistencies.** Images with 7 mm tag distance, SNR 20, and 3.5x7.0x7.0 mm 3 pixel size for varying amounts of shift increasing from left to right (values are written at the bottom right corner for each case) on short-axis (a) and long-axis (b) views. IP and TP stand for the direction of shift, in-plane and through-plane, respectively. Images represent the configuration at end-diastolic time frame. The arrows indicate the regions with the largest amount of error due to shift for IP (± 6 IP) and TP (± 6 TP).

mm to 7 mm. For the SNR analysis, three datasets are chosen to represent different levels of pixel size (2.0x4.0x4.0 mm 3, 3.0x6.0x6.0 mm 3, and 3.5x7.0x7.0 mm 3) with varying noise levels in the same interval as used for the isotropic case. The anisotropic image analysis is enriched by going beyond random errors and considering a source of systematic error (Fig 3). Since *in vivo* 3D tagged images are acquired in three successive breath-holds, individual image stacks may not be perfectly aligned [10]. Hence, we introduce shifts between individual image stacks to better reproduce the *in vivo* image acquisition, and understand the effect on deformation analysis. In this study, we simulated in-plane (IP) and through-plane (TP) shifts of image stacks with 3.5x7.0x7.0 mm 3 pixel size and SNR 20. For this purpose, the image stack tagged in Z is kept constant for both cases while the two other stacks are shifted in the respective planes by the same amount (Fig 3).

## 3 Results

### 3.1 Isotropic images

**3.1.1 Impact of TPR.** Fig 4 presents the error analysis as a function of TPR, where the black curves represent change in pixel size for a constant tag distance of 7 mm while the red curves stand for constant pixel size of 1 mm as tag distance varies. Normalized mean ± standard deviation of the displacement error norm (defined Eqs (9) and (10), reported Fig 4a) and mean ± standard deviation in Green-Lagrange strain component errors (defined Eqs (12) and (13), reported Fig 4(b)–4(d)) are used as the metrics to assess strain estimation accuracy. As shown in Fig 4a, the tracking algorithm performs well in comparison to ground truth with a displacement error of 1.1 ± 0.8% relative to the reference image set. Moreover, it still performs well for TPR higher than 4.0. The largest strain error is observed in the radial component, which is 0.1 ± 0.2 for 1 mm pixel size and 3 mm tag distance (Fig 4b).

**3.1.2 Impact of SNR.** Fig 5 presents the effect of SNR on strain measurement. For all plots, black curves stand for the best performing regularization $\beta$ while the red ones represent error in the displacement field when no regularization is applied. For the best case (Inf SNR),

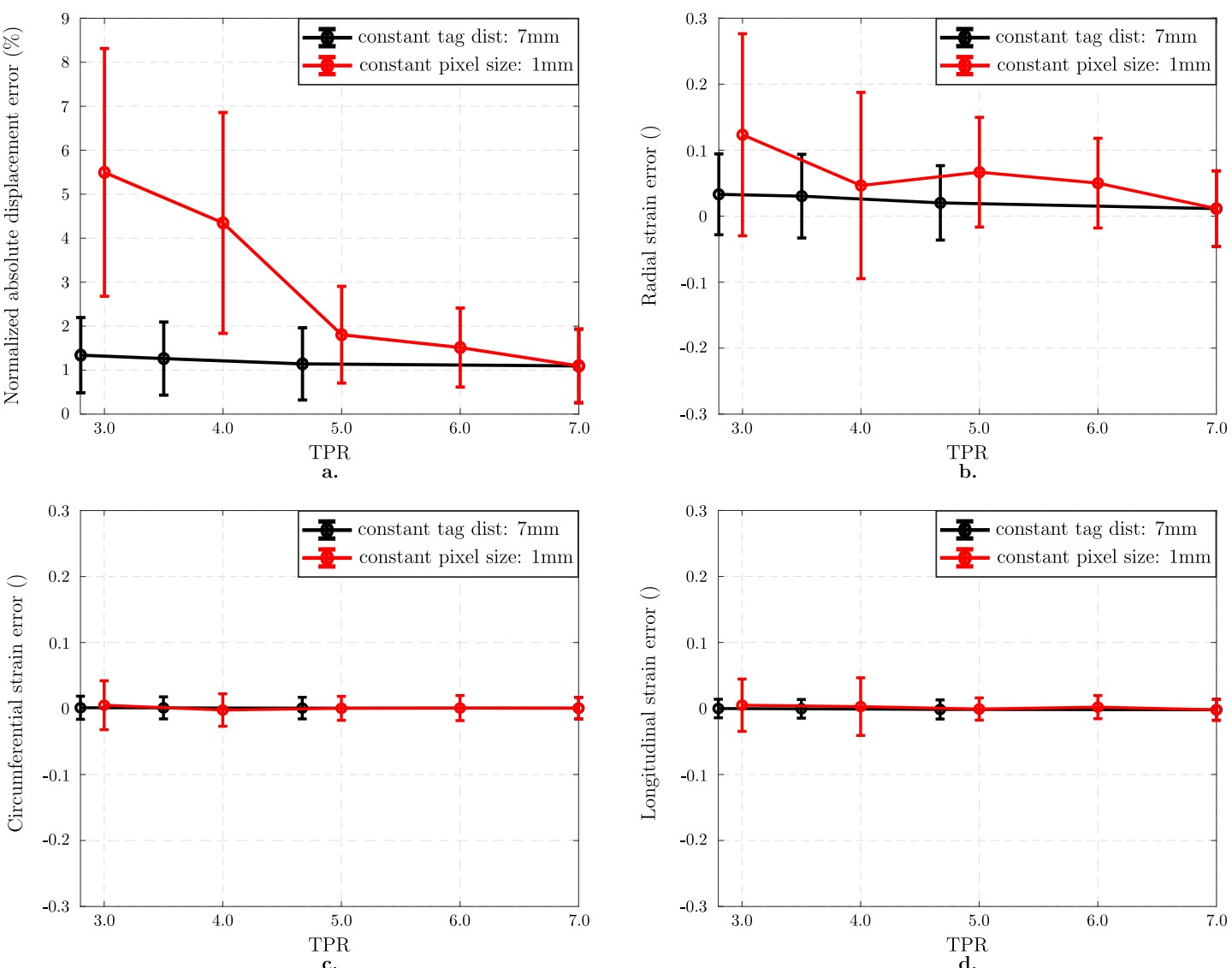

**Fig 4. Effect of Tag distance to Pixel size Ratio (TPR) on noiseless isotropic image analysis.** Normalized mean ± standard deviations in displacement error norm (%) plotted as a function of TPR (a). Black curve stands for the errors computed for the images with different pixel sizes keeping the tag distance constant at 7 mm while the red one represents the analysis results with different tag distances where the pixel size is 1 mm. Mean ± standard deviations in component-wise Green-Lagrange strain error as a function of TPR for the best performing regularization strength (b-d). Radial strain component (b) is more sensitive to a change in TPR while the circumferential (c) and the longitudinal (d) components are more accurately measured.

error in the displacement filed is 1.1 ± 0.8% (Fig 5a). For SNR 5, regularization decreases the error from 3.5 ± 2.6% (for $\beta$ = 0) to 1.6 ± 1.1% (for best $\beta$ = 0.05). A similar trend is found in Fig 5(b)–5(d) in terms of mean ± standard deviations in Green-Lagrange strain component errors. For SNR 5, errors in radial, circumferential, and longitudinal components are 0.0 ± 0.3, 0.0 ± 0.1, and 0.0 ± 0.1 when no regularization is applied. For optimal regularization, errors in these components vanishes for all components.

**3.1.3 Combined impact of TPR and SNR.** Fig 6 presents the results for the images with different isotropic resolutions varying in SNR at a fixed tag distance of 7 mm. Error in displacement field significantly increases as pixel size and SNR change from 1.0 mm to 3.5 mm

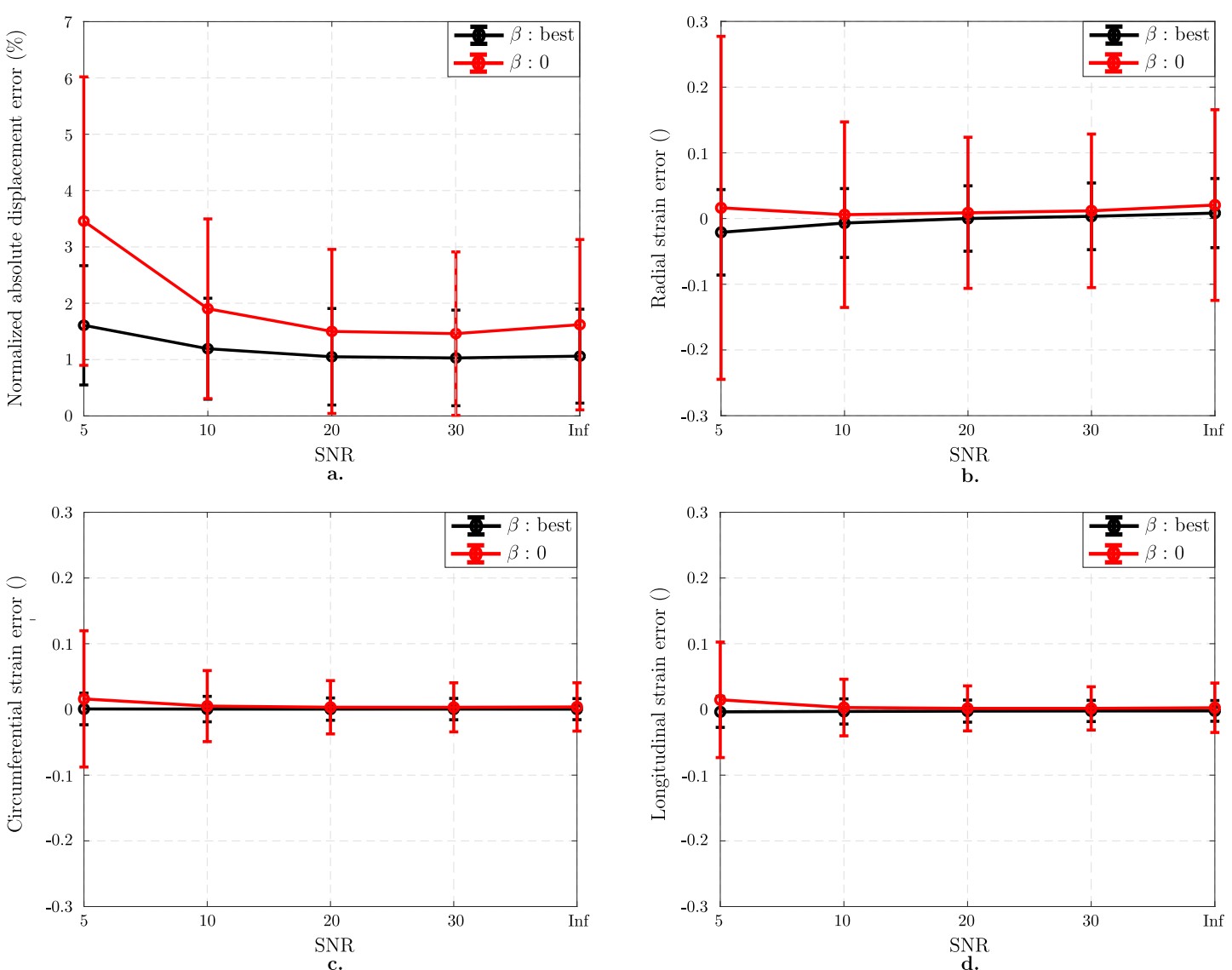

**Fig 5. Effect of SNR on isotropic image analysis.** Results on images with 1 mm pixel size and 7 mm tag distance. Normalized mean ± standard deviations in displacement error norm (%) without regularization (red curve) and with optimal regularization (black curve) (a). Regularization helps decreasing the registration error significantly. Mean ± standard deviations in component-wise Green-Lagrange strain error as a function of SNR which is independent of regularization strength for SNR≥20 (b-d).

and Inf to 5, respectively (Fig 6a). While the error in displacement field is 1.1 ± 0.8% for 1 mm pixel size and Inf SNR, it increases to 3.1 ± 1.8% when pixel size is 3.5 mm and SNR 5 (worst case). Radial strain error (Fig 6b) is larger compared to other components (Fig 6(c) and 6(d)) and 0.01 ± 0.14 for the worst case. The reader is referred to S1a Fig in S1 File for the choice of optimal $\beta$ values for varying image resolution and SNR. The two-way ANOVA analysis on SNR and image resolution shows that SNR is the main source of error in displacement field, radial and circumferential strain errors with 56%, 69% and 39%, respectively, while the corresponding contributions from image resolution are 39%, 20% and 37%. The longitudinal strain error, however, is 17% due to SNR and 72% due to image resolution (p < 0.0001).

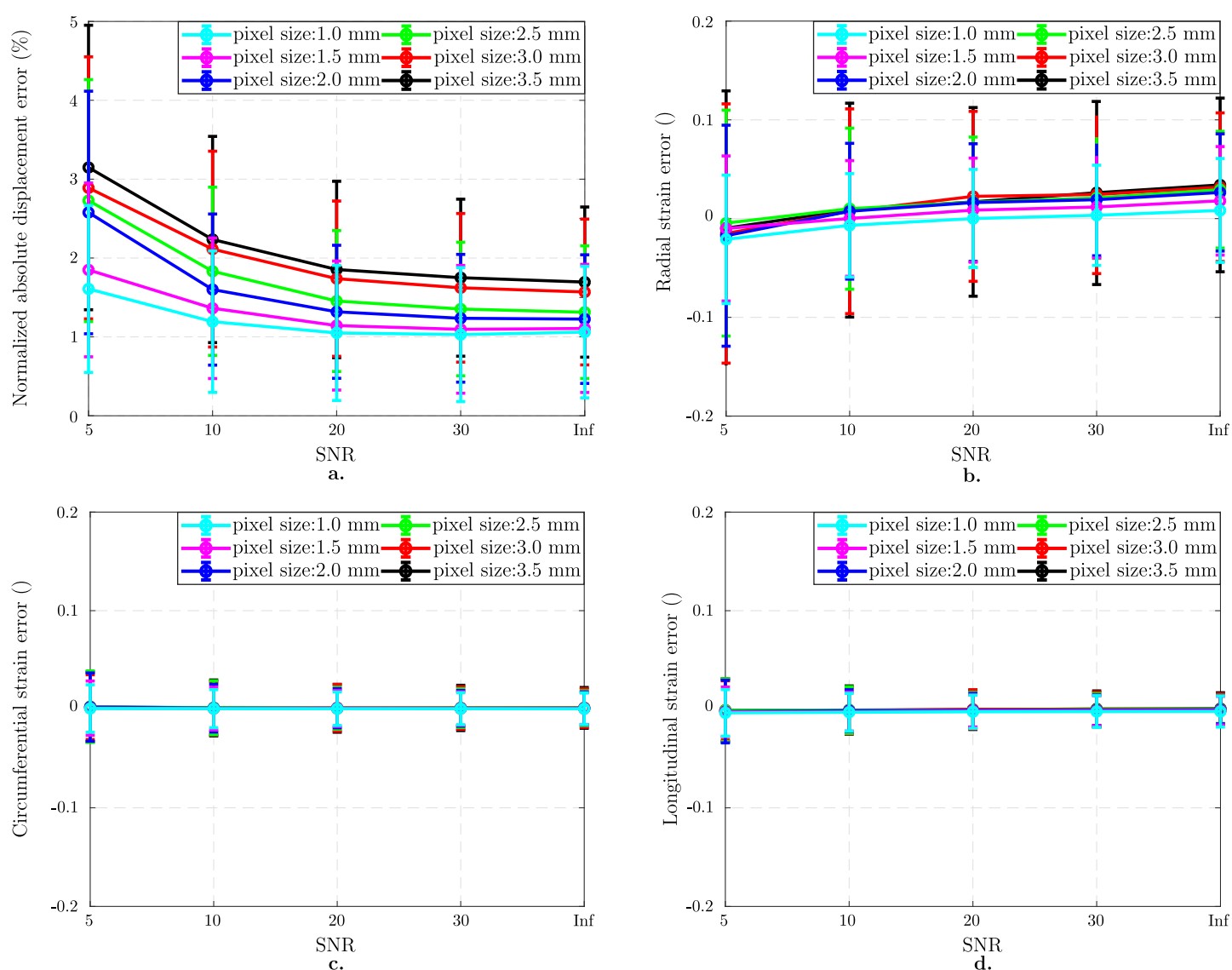

**Fig 6. Combined effect of image resolution and SNR on isotropic image analysis.** Results on images with 7 mm tag distance. Normalized mean ± standard deviations in displacement error norm (%) plotted as a function of SNR for different pixel sizes (a). Mean ± standard deviations in component-wise Green-Lagrange strain error as a function of SNR and pixel size (b-d). Combined effect of pixel size and SNR is more pronounced on the radial strain component (b) while there is almost no change in the circumferential (c) and longitudinal (d)components. Legends show the pixel size in mm.

## 3.2 Anisotropic images

**3.2.1 Impact of image resolution.** Fig 7 summarizes the analysis results on noiseless anisotropically resolved images. For each plot, the x axis represents the pixel size in tagging direction which is finer compared to the orthogonal (transverse) directions represented by the y axis. Fig 7a shows the normalized mean ± standard deviation (%) of the displacement error norm which increases from 1.5% to 2.6% as the pixel size increases from 1.5x3.5x3.5 mm 3 to 3.5x7.0x7.0 mm 3. The error in radial strain increases up to 0.06 when the pixel size in orthogonal direction is larger than 5 mm (Fig 7b). Moreover, circumferential (Fig 7c) and longitudinal (Fig 7d) components are less sensitive to change in image resolution.

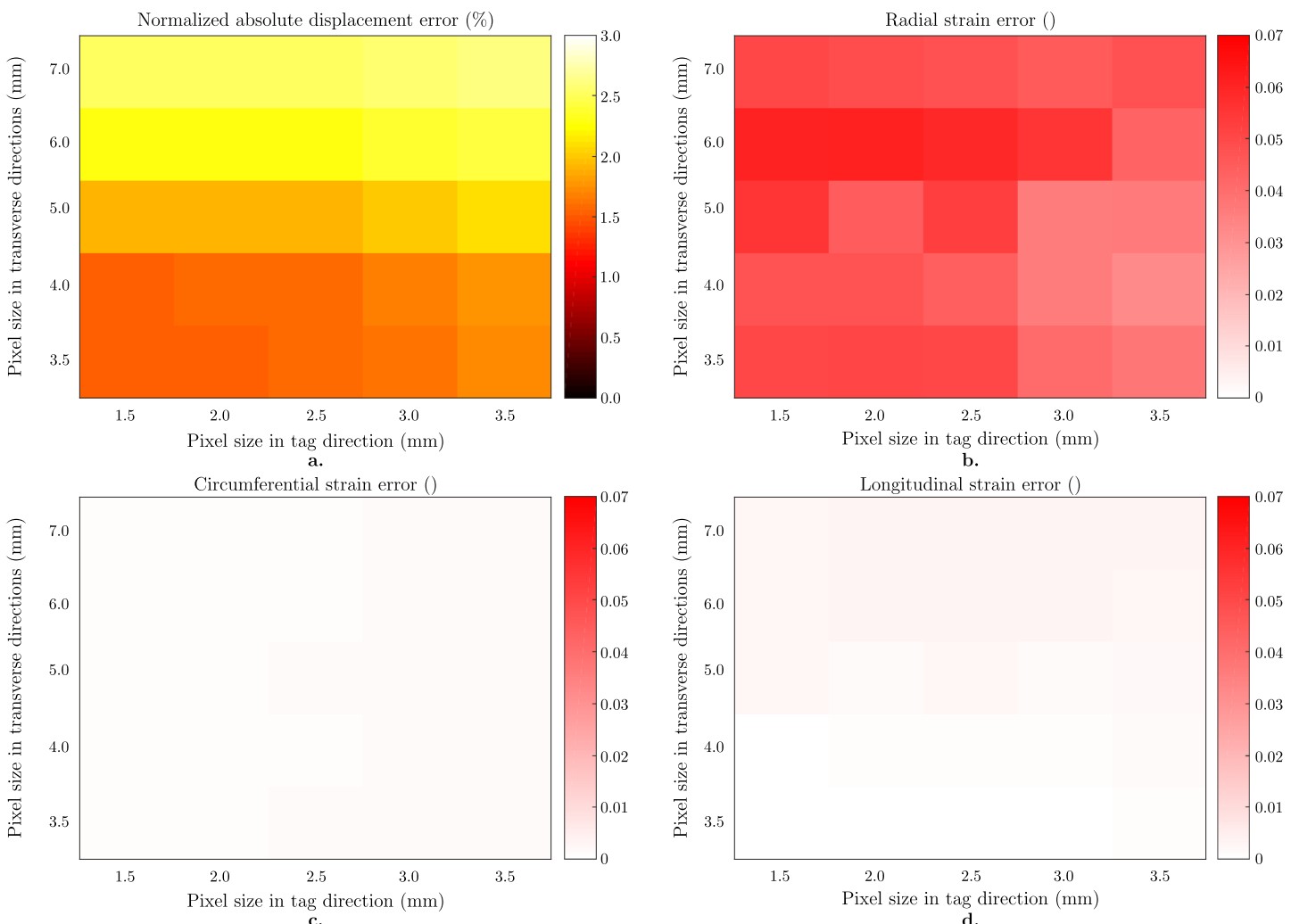

**Fig 7. Effect of image resolution on anisotropically resolved image analysis.** Results on noiseless images with 7 mm tag distance. Normalized mean ± standard deviations in displacement error norm (%) as a function of pixel size (a). Signed averages in Green-Lagrange strain component errors as a function of pixel size (b-d). Radial strain component is the most sensitive to an increase in pixel size (b), while circumferential (c) and the longitudinal (d) components are not affected at all.

**3.2.2 Combined impact of image resolution and SNR.** We observe an increase in error both in displacement field (Fig 8a) and strain components (Fig 8(b)–8(d)) for anisotropic images as SNR decreases and pixel size increases. The lowest error was found for the smallest pixel size and highest SNR: the error in displacement field is 1.5 ± 1.0% while it increases to 4.4 ± 2.8% for the worst case (3.5x7.0x7.0 mm 3 pixel size & SNR 5) (Fig 8a) The reader is referred to S1b Fig in S1 File for the choice of optimal $\beta$ values for anisotropic image resolutions (2.0x4.0x4.0 mm 3, 3.0x6.0x6.0 mm 3 and 3.5x7.0x7.0 mm 3) and varying SNR. ANOVA analysis shows that SNR is the main source of error in displacement and radial, circumferential and longitudinal strain errors with values of 62%, 78%, 64%, and 46%, respectively while image resolution contributes with 36%, 1%, 16%, and 24% to each field (for p < 0.0001).

**3.2.3 Combined impact of image resolution, SNR and geometrical inconsistencies.** Fig 9 presents the results for the images with anisotropic resolution 3.5x7.0x7.0 mm 3, tag distance of 7 mm and SNR 20 when different amounts of shift between tagged image stacks are applied. For all plots, the black curves represent the analysis results on images shifted in-plane, i.e., in

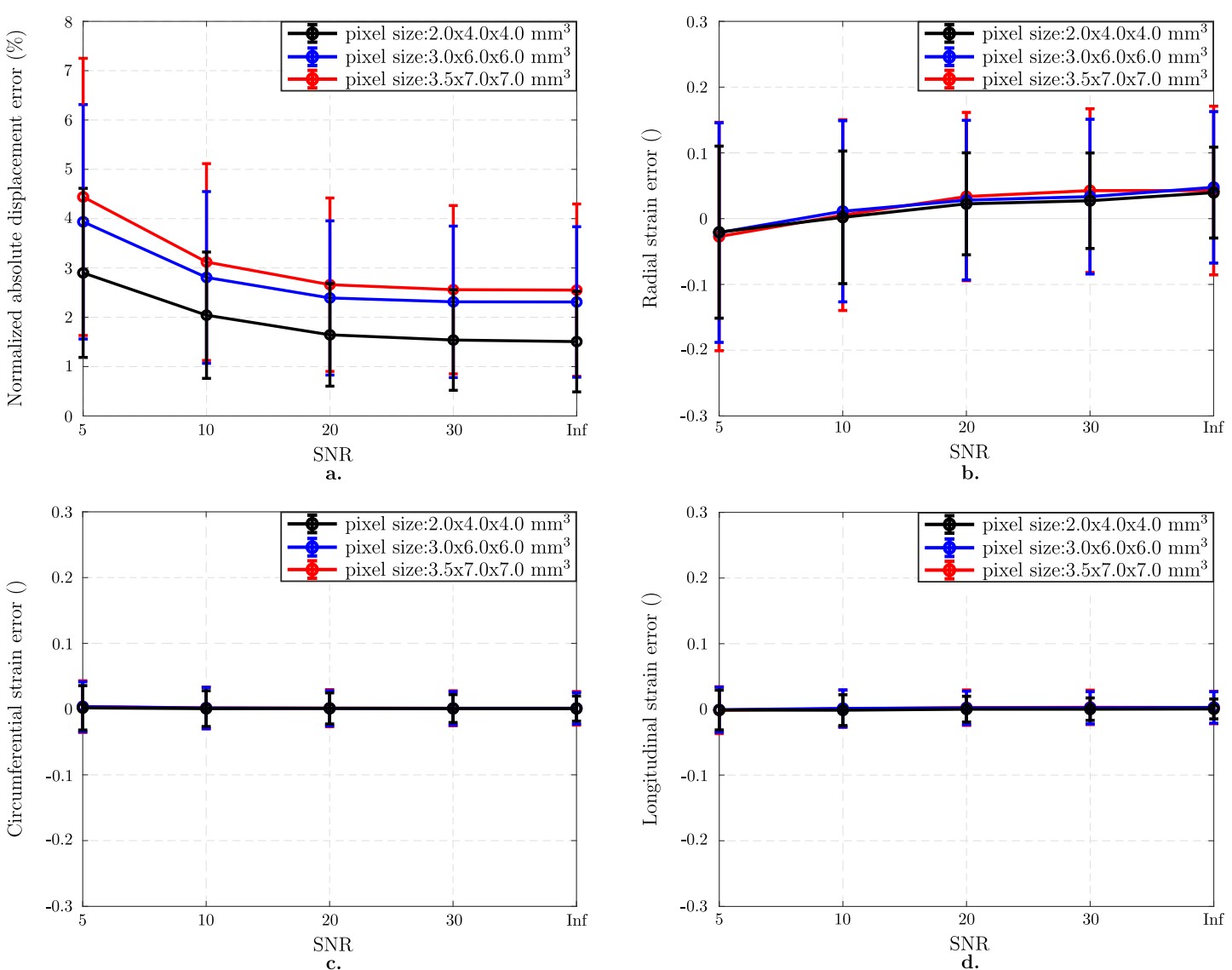

**Fig 8. Combined effect of image resolution and SNR on anisotropically resolved image analysis.** Results on images with 7 mm tag distance. Normalized mean ± standard deviations in displacement error norm (%) plotted as a function of SNR for different pixel sizes (a). Mean ± standard deviations in component-wise Green-Lagrange strain error as a function of SNR and pixel size (b-d). Combined effect of increased pixel size and decreased SNR is more pronounced on the radial strain component (b) while there is almost no change in the circumferential (c) and longitudinal (d) components.

the short-axis plane. Red curves stand for the images shifted through-plane along the long axis, where the same stacks are shifted in the transverse direction. Error in displacement field increases as the stacks are shifted further (Fig 9a). When the image stacks are shifted by ±6 mm in-plane and through plane, the error increases up to 10.1 ± 5.4% and 9.8 ± 5.0%, respectively. Effect of shifting is more significant on radial strain component (Fig 9b). The error is -0.02 ± 0.27 when the shift is ±6 mm in-plane while it is -0.01 ± 0.32 for through plane shift. However, circumferential and longitudinal strain components are quite robust to any type of shift between image stacks (Fig 9(b) and 9(c)). The localized effects of shifting are further discussed in Supporting information.

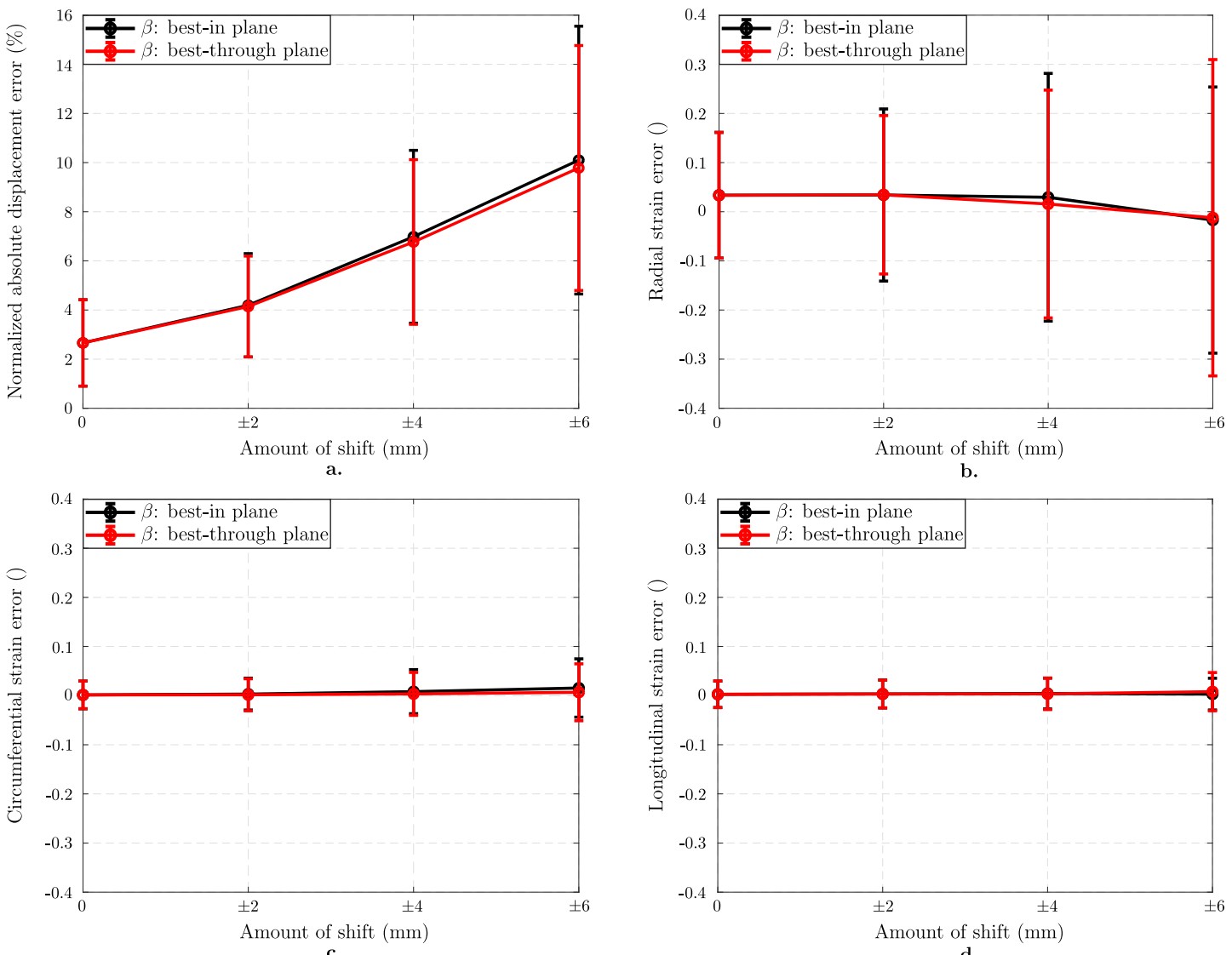

**Fig 9. Effect of geometrical inconsistencies.** Results on images with anisotropic pixel size 3.5x7.0x7.0 mm 3 and SNR 20 for 7 mm tag distance. Normalized mean ± standard deviations in displacement error norm (%) plotted as a function of in-plane and through plane shift (in mm) between image stacks (a). Mean ± standard deviations in component-wise Green-Lagrange strain error as a function of in-plane and through plane shift (in mm) between image stacks (b-d). Increasing amount of shift leads to an increased error in radial strain component (b) while there is almost no change in the circumferential (c) and longitudinal (d) components.

## 4 Discussion

We have investigated the effect of image resolution, tag distance and SNR on deformation quantification from 3D tagged images and have illustrated the challenges in quantifying radial strain.

For noiseless images, registration can be performed without regularization for all TPR values greater than 2 in case the tag distance is kept constant (see Fig 4). Although TPR is a dimensionless parameter essentially controlling the impact of tag distance and image resolution on the motion tracking, the tracking error still differs when varying TPR by changing tag distance (red curve of Fig 4) or pixel size (black curve of Fig 4). This is due to the fact that the tracking also depends on the different spatial scales of the problem, i.e., the object size, the

displacement length, and the mesh size (which defines the tracked displacement characteristic length). Indeed, for an object of given size and an image of given TPR, an image with coarser discretization will lead to more partial voluming, and thus larger tracking error. Similarly, for a given object motion, jumps across tag lines will be facilitated on images with smaller tagline distances. Note that such tag jumps are not directly caused by our choice of tracking method or temporal discretization; they are inherent to tracking with periodic tagging patterns and have been reported for other tracking methods such as harmonics phase imaging [20]. Therefore, we studied the effect of TPR in the optimal range of tag distance; coarse enough to prevent the tag line jumping during tracking and fine enough to have good image contrast. Practically, for future CMR tagging acquisitions, this suggests using rather large tag distances ($\geq$ 5 mm) to prevent tag jumping when tracking cardiac motion. Focusing on results with 7 mm tag distances (black curve of Fig 4), tracking results are not drastically influenced by TPR, i.e., by image resolution. This holds true for the displacement error, as well as for the strain error, in terms of both mean and standard deviation. Radial strain error is generally larger than circumferential and longitudinal strain errors; however, this higher sensitivity of radial strain is not significantly affected by image resolution.

Conversely, for noisy images, SNR has a significant impact on tracking errors (see Fig 5), especially when tracking without regularization (red curve of Fig 5). Indeed, in this case, the standard deviation of the strain error explodes for low SNR. This is true for all components, though it is more significant for radial strain compared to circumferential and longitudinal strains. However, the higher error of radial strain can be alleviated by the use of regularization (black curve of Fig 5). Indeed, with regularization, even though the radial strain error standard deviation is higher than for circumferential and longitudinal strains, the tracking error is not influenced by SNR.

Looking at the combined effect of image resolution and SNR (see Fig 6), we observed a considerable increase in displacement error with increasing pixel size and decreasing SNR. This trend is seen despite the use of optimal regularization and fine temporal discretization, which suggests that the cause may be found in the image properties, i.e., spatial resolution and SNR. More specifically, we found that radial strain was much more sensitive to image characteristics than circumferential and longitudinal components. This is in line with observation made on tagged CMR image processing [18, 26]. The elevated error in the radial component can also be associated to the thickness of the myocardial wall, which is much smaller than the circumference and length of the ventricle [44]. However, in this study, we did not investigate the effect of wall thickness on motion quantification in detail. Instead, we limited the scope of the study to the effect of imaging parameters only, fixing the geometrical and material biomechanical model parameters to normal values. One solution would be to increase the tag line density in the radial direction; however, as discussed previously, this would require to make the tracking more robust with respect to tag jumping. Alternatively, for future CMR acquisitions, tagged image resolution needs to be improved for better radial strain mapping.

The effect of SNR is more pronounced for the estimation of the displacement field and radial strain when combined with the anisotropic voxel sizes (see Figs 7 and 8). Although the anisotropic images represent a better approximation of the in-vivo imaging setting, we do not observe a dramatic increase in error for low SNR values by the choice of optimal $\beta$ (Fig 8a and 8b). On the contrary, circumferential and longitudinal strains did not show significant increase in error when choosing an anisotropic voxels size within the range evaluated in this study. And again, since this is found with a fine temporal discretization and an optimal motion tracking method, this reinforces the point that reduced radial strain measured on *in vivo* images is induced by limited image spatial resolution and SNR.

To further study this point and mimic multi breath hold in vivo acquisitions, we investigated geometrical stack misalignment as additional source of error (see Fig 9). While the direction of shifting appeared to not influence the error, we observed an increased error more pronounced in the radial strain component while the other two components remain insensitive when stacks are shifted further apart.

Radial strain quantification remains sensitive to low SNR, TPR and slice misalignment, however the quantification of circumferential and longitudinal strains appears to be affected only little across the parameter range investigated in this study. Both global longitudinal strain (GLS) and global circumferential strain (GCS) have been used as prognostic value for disease severity in the clinical setting [45, 46], with GLS being the most widely used metric to predict heart failure [47]. To this end, the tracking method used in this study [26], offers a robust way for the assessment of clinically used strain values.

In order to make radial strain measurement more robust and potentially useable in the clinical setting, multiple attempts have already been made. Combining untagged and tagged images has been shown to perform better compared to using only one type of image [18, 48], though the use of untagged images suggests that strain heterogeneities might not be mapped accurately. Proposed as a new imaging modality, subtly tagged steady state free precession (SubTag SSFP) feature tracking has allowed for acquiring regional deformation and ventricular function in a single MRI scan, hence, achieving shorter scan time and better mechanical assessment [49]. Lastly, the imaging process itself has also been modeled to understand the effect of partial voluming, which deteriorates the image quality, and as a consequence, deformation assessment [50]. Nevertheless, our study suggests that by improving image resolution, measuring accurate radial strain fields could be achieved based solely on tagged MRI with existing image processing techniques.

## 5 Conclusion

In this study, we systematically investigated the accuracy and precision of deriving LV strain components from CMR tagged images by finite element digital image correlation. It shows the robustness of GLS and GCS to varying image characteristics, supporting their reliability and common usage in the clinical setting. Moreover, the error analyses suggest that it is worthwhile investing into higher spatial resolution when planning tagged CMR acquisitions in order to obtain a robust radial strain estimate.

## Supporting information

**S1 File.**
(DOCX)

## Author Contributions

**Supervision:** Christian T. Stoeck, Sebastian Kozerke, Martin Genet.

**Writing – original draft:** Ezgi Berberoğlu.

**Writing – review & editing:** Christian T. Stoeck, Philippe Moireau, Sebastian Kozerke, Martin Genet.

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
