## [Decision Letter · Decision Letter 0]

24 May 2021

PONE-D-21-06747

In-silico study of accuracy and precision of left-ventricular strain quantification from 3D tagged MRI

PLOS ONE

Dear Dr. Genet,

I am the new AE for this paper.  It seems one review indicated that this version is a revision.  Anyway,  this is the first time Reviewer 2 saw this.  Please read the comments and respond carefully.

Thank you for submitting your manuscript to PLOS ONE. After careful consideration, we feel that it has merit but does not fully meet PLOS ONE’s publication criteria as it currently stands. Therefore, we invite you to submit a revised version of the manuscript that addresses the points raised during the review process.

We look forward to receiving your revised manuscript.

Kind regards,

Dalin Tang

Academic Editor

PLOS ONE

Journal Requirements:

Reviewers' comments:

Reviewer's Responses to Questions

**Comments to the Author**

1. Is the manuscript technically sound, and do the data support the conclusions?

Reviewer #1: Yes

Reviewer #2: Yes

2. Has the statistical analysis been performed appropriately and rigorously? 

Reviewer #1: Yes

Reviewer #2: No

3. Have the authors made all data underlying the findings in their manuscript fully available?

Reviewer #1: Yes

Reviewer #2: Yes

4. Is the manuscript presented in an intelligible fashion and written in standard English?

Reviewer #1: Yes

Reviewer #2: Yes

5. Review Comments to the Author

Reviewer #1: The authors did a very good job of revision that greatly improved the overall quality of this manuscript.

They answered all my questions point by point and they are to be congratulated for the quality of this revision.

This work seems to me sufficiently improved to be published.

Reviewer #2: The present manuscript evaluates a finite-element method for strain quantization using an in-silico phantom. The approach of using an in-silico phantom is likely of interest to investigators in the field and the analysis of how parameters like tag distance and pixel size affect the accuracy of the strain quantification is important. The manuscript does not open completely new ground in the field, but it could make a useful contribution to the field, and the results are important for expanding our current knowledge about the limits of strain quantification with current MRI methods and post-processing techniques. This comment is also not meant to the play down the indisputable technical sophistication of the work. The manuscript could be improved by avoiding some errors like the use of terms or abbreviations before they are properly defined, providing some more information about the in-silico phantom, and in general helping the reader gain an easier intuitive understanding of the results and the fundamental limits to reducing strain error.

1) In-silico phantom: it would be useful to have some basic information about the phantom in terms of ES and ED cavity volume, mean wall thickness at ED and ES. In particular the relation between mean wall thickness and tag-line spacing is of some relevance for understanding when increased tag-line distance impacts on the accuracy of radial strain estimation. In fact, in the case of radial strains it may be useful to not only assess strain quantification as a function of TPR but also as a function of the tag-line-distance to mean wall-thickness ratio. It should be made clearer to the reader that for radial strain quantification it is important that at least two intersection points/lines fall within the wall thickness when moving in the radial direction. This limitation is much less of a constraint for the circumferential and longitudinal components.

2) Page 4: it may be better to use different symbols for the functional J in equation 6 and the quantity J in equation 7, as they apparently represent different things.

3) Equation 4: Generally, with regularization the primary quantity that is to be minimized - in this case the similarity metric - is not weighted by the regularization parameter. Why did the authors choose in this case to use a (1-beta) factor to weigh the similarity metric, in addition to the customary weighing of the regularization term by beta? Is there any logic behind requiring that the two weighting factors to sum to one, i.e. choosing weighting factors of (1-beta) and beta?

4) Page 4, choice of regularization parameter value: for practical application of the proposed method the regularization parameter would have to be chosen without recourse to the ground truth, which in practice means using some heuristic like the L-curve criterion or similar. Could the authors please comment on this and also relate this to the question why they weighed both terms in equation 6 by (1-beta) and beta, respectively. How does this affect the choice of optimal regularization parameter when the ground truth is not known?

5) TPR (Tag distance to Pixel size Ratio) is first defined in Results, but the abbreviation is used beforehand in Methods. The authors should define new symbols or abbreviation at first use in the manuscript.

6) The nature of the shifts that were applied to analyze the effects of varying degrees of geometric inconsistencies are not clear. In what direction where shifts applied to the images? Would a shift applied in one direction not result in different degrees of accuracy depending on the location of a myocardial segment relative to the center of the LV and the direction of the shift? This is rather confusing, and the authors need to make a better effort in explaining their method of evaluating geometric inconsistencies!

7) Could the authors indicate in which cases (figures) the change of strain parameter or error is significant? It looks like in many cases the change of one parameter (e.g. SNR in the case of Figure 8) has little effect on the quantification error! I think that indicating which changes are statistically significant is better than just speaking of “a considerable increase” of quantity X or similar.

8) The figures are appended in the manuscript PDF in reverse order (in terms of the numbering).

9) Equation 1: The image-related quantity X_i is not explained.

6. PLOS authors have the option to publish the peer review history of their article (what does this mean?). If published, this will include your full peer review and any attached files.

Reviewer #1: No

Reviewer #2: **Yes: **Michael Jerosch-Herold

---

## [Author Response · Author response to Decision Letter 0]

29 Jul 2021

We would like to thank both reviewers for carefully reading our manuscript, and making constructive comments. We have addressed all reviewers’ comments, as detailed in our point by point response and revised manuscript.

---

## [Decision Letter · Decision Letter 1]

25 Aug 2021

PONE-D-21-06747R1

In-silico study of accuracy and precision of left-ventricular strain quantification from 3D tagged MRI

PLOS ONE

Dear Dr. Genet,

Thank you for submitting your manuscript to PLOS ONE. After careful consideration, we feel that it has merit but does not fully meet PLOS ONE’s publication criteria as it currently stands. Therefore, we invite you to submit a revised version of the manuscript that addresses the points raised during the review process.

We look forward to receiving your revised manuscript.

Kind regards,

Dalin Tang

Academic Editor

PLOS ONE

Journal Requirements:

Reviewers' comments:

Reviewer's Responses to Questions

**Comments to the Author**

1. If the authors have adequately addressed your comments raised in a previous round of review and you feel that this manuscript is now acceptable for publication, you may indicate that here to bypass the “Comments to the Author” section, enter your conflict of interest statement in the “Confidential to Editor” section, and submit your "Accept" recommendation.

Reviewer #2: All comments have been addressed

2. Is the manuscript technically sound, and do the data support the conclusions?

Reviewer #2: Yes

3. Has the statistical analysis been performed appropriately and rigorously? 

Reviewer #2: Yes

4. Have the authors made all data underlying the findings in their manuscript fully available?

Reviewer #2: Yes

5. Is the manuscript presented in an intelligible fashion and written in standard English?

Reviewer #2: Yes

6. Review Comments to the Author

Reviewer #2: The manuscript has been skillfully revised by the authors. After examining the revised manuscript I still have two queries for the authors, but these should only require very minor changes to the manuscript.

1) Equation 1 implies a very simply preparation for spatial modulation of magnetization, while in practice e.g. a binomial scheme is going to result in sharper tag lines. The authors could point out that in this sense there are potentially looking at a worst case scenario in terms of tag line contrast.

2) The authors state in the legend for the new figure S1: "The tracking algorithm requires a higher regularization strength as image resolution and SNR decrease." Could the authors clarify if image resolution causes an increase of the regularization weight independent of the SNR, or if the effect of image resolution is through the natural decrease of SNR? It would make sense that higher image resolution does require stronger regularization, as the higher image resolution is likely increasing how ill-posed the inverse problem is. I am also not sure if what they call a "decrease" of image resolution is correct terminology, because if pixel size decreases I would call this "higher resolution" and not a decrease of resolution. The authors should consult with a native speaker.

7. PLOS authors have the option to publish the peer review history of their article (what does this mean?). If published, this will include your full peer review and any attached files.

Reviewer #2: **Yes: **Michael Jerosch-Herold

---

## [Author Response · Author response to Decision Letter 1]

4 Oct 2021

We have improved the revised manuscript considering the reviewer’s comments carefully.

---

## [Editor Report · Decision Letter 2]

11 Oct 2021

In-silico study of accuracy and precision of left-ventricular strain quantification from 3D tagged MRI

PONE-D-21-06747R2

Dear Dr. Genet,

We’re pleased to inform you that your manuscript has been judged scientifically suitable for publication and will be formally accepted for publication once it meets all outstanding technical requirements.

Kind regards,

Dalin Tang

Academic Editor

PLOS ONE
---

## [Editor Report · Acceptance letter]

15 Oct 2021

PONE-D-21-06747R2 

In-silico study of accuracy and precision of left-ventricular strain quantification from 3D tagged MRI 

Dear Dr. Genet:

I'm pleased to inform you that your manuscript has been deemed suitable for publication in PLOS ONE. Congratulations! Your manuscript is now with our production department. 

Kind regards, 

on behalf of

Professor Dalin Tang 

Academic Editor

PLOS ONE